# Trust, Interaction with Neighbors, and Vaccination during the COVID-19 Pandemic: A Cross-Sectional Analysis of Chinese Data

**DOI:** 10.3390/vaccines11081332

**Published:** 2023-08-06

**Authors:** Takashi Oshio, Ruru Ping

**Affiliations:** 1Institute of Economic Research, Hitotsubashi University, Tokyo 186-8601, Japan; 2Graduate School of Economics, Hitotsubashi University, Tokyo 186-8601, Japan; ed215104@g.hit-u.ac.jp

**Keywords:** China, interaction, multilevel analysis, trust, vaccination

## Abstract

The COVID-19 pandemic significantly impacted public health and quality of life, leading to government recommendations for vaccination. Using cross-sectional data from a nationwide population-based survey conducted in China (*N* = 6860), this study aimed to examine the associations between individual vaccine uptake and general trust in others, trust in government, and interaction with neighbors. We conducted a multilevel logistic regression analysis to examine the relevance of these factors at the individual and community levels. Among young adults, higher levels of general trust at both levels were positively associated with vaccination, with odds ratios (OR) of 1.35 (95% confidence interval [CI]: 1.07, 1.70) and 1.58 (95% CI: 1.14, 2.18), respectively. We also found a positive association between vaccination and community-level interaction with neighbors, with ORs of 1.55 (95% CI: 1.11, 2.17). In contrast, among older individuals, vaccination was positively associated only with individual-level interaction with neighbors, with an OR of 1.55 (95% CI: 1.15, 2.08). The results indicated that vaccine uptake was associated with an individual’s views of society and the social environment of the community, with substantial variations between the young and the old. Our findings emphasize the significance of public health measures to strengthen neighborhood interactions among older adults.

## 1. Introduction

The COVID-19 pandemic has had a serious impact on public health and quality of life, leading governments in many countries to recommend vaccination. Several studies have demonstrated that vaccine uptake as well as vaccination willingness or hesitancy are associated with various factors, including demographic attributes (e.g., age, sex, and marital status), socioeconomic background (e.g., educational attainment, income, and working status), health-related behaviors (e.g., regular influenza vaccination and medical checkups), and concerns about the safety and effectiveness of vaccines [1,2,3,4,5,6,7,8,9].

In addition to these key factors, an individual’s views and attitudes towards society have been highlighted as correlates of vaccination in an increasing number of studies. Notably, higher levels of social capital [10,11,12] and trust in government [13,14,15] are found to be in favor of vaccination in many countries.

Building upon previous studies [10,11,12,13,14,15], we examined how general trust in others, trust in government, and interaction with neighbors affect an individual’s decisions on vaccine uptake. General trust in others is a key element of social capital, which is defined as the quality of relationships among community members [16]. Previous studies often use general trust as an indicator of social capital to examine its association with vaccine hesitancy [10,11,12]. Higher general trust is expected to increase individuals’ concern for others and engagement in practices that aim to improve the overall situation.

Trust in government, which refers to people’s beliefs that the government acts transparently and fairly in accordance with the public interest, has also been investigated as a factor related to vaccination willingness [13,14,15]. Analyzing vaccination practices in the context where the government takes the lead in managing the pandemic crisis makes it even more important to consider trust in the government.

As a correlate of vaccination, we also focused on interaction with neighbors, which is considered a vital component of neighborhood-based social capital [17,18]. More frequent and close interactions with neighbors are expected to help individuals share pandemic-related information and increase support for neighbors, thereby promoting vaccine uptake.

Unlike most previous studies [11,12,13,14,15], this study distinguished individual- and community-level measures of general trust, trust in government, and interaction with neighbors, and examined their associations with vaccination at both levels. Researchers studying social capital have demonstrated that health outcomes and subjective well-being are correlated with social capital at both individual and community (neighborhood) levels [19,20,21,22,23]. However, regarding COVID-19 vaccination, most studies have primarily focused on individual-level social capital, mainly in terms of general trust [10,11,12]. Recently, some scholars have suggested that aggregate social capital may influence residents’ responsiveness to pandemic shocks [24,25]. Nevertheless, whether residing in a community with high trust in government or interaction with neighbors is correlated with vaccination, independent of individual-level factors, remains to be addressed. Our study aimed to fill in this empirical gap.

By the end of 2022, the proportion of people in China who received at least one dose of the COVID-19 vaccine stood at 92%, surpassing the rates of 79% for high-income countries and 69% for the global population [26]. As one of the major producers of COVID-19 vaccines, China had the capacity to provide an ample number of vaccine doses for its domestic population. Considering the fact that vaccination is not mandatory, it is intriguing to explore the extent to which the exceptionally high vaccination rate can be attributed to people’s attitudes towards society or the government.

Using cross-sectional data from a nationwide, population-based social survey conducted in China between late June and the end of September 2021, we examined the associations between vaccine uptake with general trust, trust in government, and interaction with neighbors at both individual and community levels. Additionally, we investigated how the association between vaccine uptake and general trust, trust in government, and interaction with neighbors differed between young and older populations. This investigation was prompted by our observation of a low vaccination rate among older respondents in our study sample, which is consistent with the findings from other studies [27]. Our findings regarding people’s relationships with and attitudes toward society can provide valuable insights for public health authorities to better design and manage immunization campaigns in the future, particularly when faced with new public health emergencies.

## 2. Materials and Methods

### 2.1. Study Sample

We used data from the 2021 Chinese General Social Survey (CGSS). Initiated in 2003, the CGSS was the first nationwide and continuous large-scale social survey project in China. The CGSS aims to collect data on family structure, education, employment, health, social attitudes, and other important topics in order to monitor changes in Chinese society over time and investigate social issues with theoretical and practical significance. The China Survey and Data Center of Renmin University of China is responsible for the implementation, management, and data release of the project. The CGSS adopted a multi-stage stratified sampling approach. In CGSS, counties served as primary sampling units, followed by urban communities and rural villages as secondary sampling units, and households were randomly selected using a mapping sampling approach [28].

The CGSS has conducted 15 waves between 2003 and 2022. The 2021 survey, the 14th wave of the CGSS, covered 19 province-level administrative regions (referred to as “provinces” hereafter) in China. A total of 320 communities (villages (Cun) and neighborhood committees (Ju wei hui)) were selected, with approximately twenty-five households chosen from each community. For each selected household, one individual aged 18 years or above was interviewed.

A total of 8148 participants responded to the core modules of the 2021 survey. To focus on individuals’ vaccination decisions, we excluded 1288 participants who did not meet the vaccination requirements and did not receive vaccination. The CGSS questionnaire asked about the reasons for non-vaccination, which included an option of not meeting vaccination requirements. We used this information as an exclusion criterion for our study sample. As a result, our analysis focused on 6860 individuals (3730 men and 3130 women). Almost all participants completed the vaccination program at the time of the survey, with 94.5% receiving the vaccine between March and July 2021. For this study, we used publicly available data from the Chinese National Survey Data Archive [29]; therefore, no additional ethical approval was required.

### 2.2. Measures

#### 2.2.1. Outcome and Key Independent Variables

As an outcome, we focused on whether individuals were vaccinated before the survey time and constructed a binary variable to indicate vaccination by allocating a value of one to those who reported receiving a vaccination and zero to those who did not. There were no questions regarding the number of times individuals had been vaccinated.

Three key independent variables were considered to explain vaccination outcomes: general trust, trust in government, and interaction with neighbors. We constructed a binary variable for each construct because it is difficult to assess it by a continuous measure. Moreover, a binary variable can help straightforwardly assess the magnitude of its association with vaccine uptake.

Regarding general trust, participants were asked to respond to the question, “In general, do you agree that most people can be trusted in this society?”, with the following options: strongly disagree, disagree, cannot say I agree or disagree, agree, strongly agree, and don’t know. We constructed a binary variable for general trust by allocating a value of one to the answers strongly agree or agree and zero to the other responses.

Regarding trust in government, no question in CGSS directly asked about it; instead, the participants were asked to respond to the question, “During the severe pandemic, do you think the Chinese government has the right to do the following things? a. close a business or workplace; b. require people to stay at home; c. monitor and track infected persons through digital devices (such as mobile phones); d. require people to wear masks; e. prohibit public gatherings; f. isolate the infected person; g. temporarily close primary and secondary schools and kindergartens; and h. isolate the infected person”, with the following options: definitely do, probably do, probably do not, definitely do not, cannot choose, and refuse to answer to the question. Given that a substantial portion of the participants answered definitely do for each of these eight items, we constructed a binary variable for trust in government by allocating a value of one to those who answered definitely do to all eight items and zero to others.

As for interaction with neighbors, the participants were asked to respond to the question, “How often do you have social entertainment activities with your neighbors (such as visiting each other, watching TV together, eating, playing cards, etc.)?”, with the following options: almost every day, once or twice a week, several times a month, about once a month, several times a year, once a year or less, never, and don’t know. We constructed a binary variable for interaction with neighbors by allocating a value of one to those who answered almost every day, once or twice a week, or several times a month and zero to others. 

#### 2.2.2. Covariates

As for individual covariates, we considered variables for each category of sex, age (below 30, 30–39, 40–49, 50–59, 60–69, 70–79, and 80+), marital status (married, unmarried, and divorced/separated), living alone, educational attainment (illiterate, primary school, junior high school, high school, and college or above), occupational type (no work, farming, government-related work, private or foreign company, self-employed, and other), family income level (low, middle, high, and unanswered), having poor self-rated health, belonging to agricultural hukou, and being a Communist Party member. Family income was adjusted for family size by dividing it by the square root of the number of family members and then dividing it into tertiles. For poor self-rated health, we constructed a variable by allocating a value of one to those who chose very unhealthy or relatively unhealthy among the options of very unhealthy, relatively unhealthy, average, relatively healthy, and very unhealthy in response to the question, ‘What do you think your current health status is?’

We also consider two covariates at the provincial level. First, we captured the severity of COVID-19 infections. To this end, we collected the total number of new COVID-19 cases between January 2020 and December 2021 for each province from the National Health Commission’s website [30] and obtained the population size data for each province from the China Statistical Yearbook of 2021 and 2022. We then computed the number of new infection cases per capita for each province. Subsequently, we constructed variables for high, middle, and low levels of new cases per capita, corresponding to <1, 1–4, and >4 cases per million, respectively, based on their actual distribution. Second, we constructed binary variables for each tertile (low, middle, and high) of province-level vaccination rates. These rates were computed from the CGSS sample and were considered to reflect the status of public health infrastructure at the provincial level [25].

### 2.3. Statistical Analysis

We started with a descriptive analysis, in which we described the distribution of the vaccination rates by sex and age and compared the vaccination rates to high and low levels of general trust, trust in government, and interaction with neighbors, without controlling for any covariate.

For the regression analysis, we first created variables for community-level general trust, trust in government, and interaction with neighbors by applying the econometric method suggested in previous studies [20,21,31,32]. For instance, regarding general trust, we first estimated a linear regression model that included multilevel fixed effects to explain a binary variable representing the general trust of individual *i* residing in community *j* of province *k*:(1)$$GT_{ijk}=\alpha +\sum_{m}\beta_{m}x_{ijk}+e_{1i}+e_{2j}+e_{3k}+\varepsilon_{ijk,}$$
where $GT_{ijk}$ denotes the general trust variable; α represents the overall mean of general trust; *x* is a vector of individual covariates (sex, age, marital status, living alone, educational attainment, occupation, family income level, poor self-rated health, hukou type, and being a Communist Party member); *e*_1_, *e*_2_, and *e*_3_ are individual-, community-, and province-level fixed effects, respectively; ε represents the error term. This approach aimed to capture the components of the general trust attributable to each community while controlling for individual- and province-specific factors. The key parameter is the community-level fixed effect, *e*_2_, which indicates the extent to which the general trust in community *j* differs from the overall mean of general trust, *α*. Therefore, *e*_2_ is considered a measure of community-level general trust, where higher (lower) values indicate higher (lower) levels of community-level general trust. We constructed a binary variable of high community-level general trust by allocating a value of one to *e*_2_ > 0, indicating that the community fixed effect was above the average (weighted by the number of respondents in each community), and zero otherwise. Similarly, we constructed binary variables for high community-level trust in the government and interaction with neighbors.

Using these community-level measures, we estimated three multilevel logistic regression models (Models 1–3) to examine the likelihood of vaccination in men and women separately. These models encompassed three levels (province, community, and individual) and featured random intercepts at the provincial and community levels. In the case of general trust, Model 1 used individual-level general trust as the key independent variable along with covariates. Model 2 substituted individual-level general trust with community-level general trust, and Model 3 included both individual- and community-level general trust. We estimated Models 1–3 for the trust in government and interaction with neighbors.

## 3. Results

Table 1 summarizes key features of the study sample, and Table 2 reports the vaccination rates by sex and age. Among the entire sample, 89.2% of men and 84.8% of women were vaccinated by the time of the survey. The vaccination rates were lower among individuals 60 years of age or older compared to the younger groups for both men and women. Table 3 compares the vaccination rates among individuals with high and low levels of general trust, trust in government, and interaction with neighbors. The differences in rates were tested using Welch’s *t*-test.

As shown in this table, there were significant differences in vaccination rates between the population, but not in relation to trust in government. Among older individuals, vaccination rates differed between groups of high and low trust in government, as well as groups of high and low interaction with neighbors, but not with respect to general trust. Appendix A reports the estimation results obtained from the linear regression models with multiple-level fixed effects (see Equation (1) in the case of general trust). Using the community-level binary variables for high levels of general trust, trust in government, and interaction with neighbors calculated from these multilevel fixed effects models, we employed three types of multilevel logistic models, Models 1–3, to explain the probability of vaccination.

Table 4 presents the estimated associations of trust and interaction with neighbors with vaccination in terms of odds ratios (OR), obtained from Models 1–3. Among the young population, vaccination was not associated with trust in government at either the individual or community level, although the result on individual-level trust in Model 3 showed marginal significance. However, their vaccination was positively associated with both individual-level and community-level general trust, with an OR of 1.35 (95% confidence interval [CI]: 1.07, 1.70) and 1.58 (95% CI: 1.14, 2.18), respectively in Model 3. The ORs in Model 3 were only modestly lower than those in Model 1 (OR: 1.41, 95% CI: 1.13, 1.77) and in Model 2 (OR: 1.68, 95% CI: 1.21, 2.31), suggesting that general trust at both the individual and community levels was independently associated with vaccination. Vaccination was positively related to community-level interaction with neighbors, with an OR of 1.59 (95% CI: 1.14, 2.21) in Model 2 and 1.55 (95% CI: 1.11, 2.17) in Model 3, but had no association with individual-level interaction with neighbors.

The results for the older population differed substantially from those for the young population. Vaccination among older adults was positively associated with individual-level interaction with neighbors, with an OR of 1.63 (95% CI: 1.22, 2.17) in Model 1 and 1.55 (95% CI: 1.15, 2.08) in Model 3. Although a positive association between vaccination and community-level interaction with neighbors was observed in Model 2 with an OR of 1.63 (95% CI: 1.08, 2.46), the relationship was insignificant in Model 3. Similar to the young population, vaccination was not associated with trust in the government. However, contrary to the young population, vaccination was also not associated with general trust.

The detailed results of Model 3 for general trust are presented in Appendix A. We observed that older individuals tended to avoid vaccination, which is consistent with the observations shown in Table 2. We also found that vaccination was highly associated with Communist Party membership among the young and the old and government-related occupations among young adults, suggesting that social groups closely related to the government were highly committed to vaccination. At the provincial level, per-capita infection was not associated with vaccination, whereas higher vaccination rates promoted individual vaccination. The estimation results for the covariates were almost the same across outcomes and models.

## 4. Discussion

We investigated whether vaccination uptake was associated with an individual’s relationship with and attitudes toward society using cross-sectional data obtained from a nationwide population-based social survey in China. We focused on general trust, trust in government, and interaction with neighbors, and unlike previous studies [11,12,13,14,15], we distinguished these variables at the individual and community levels and compared their associations with vaccination. In the regression analysis, we conducted multilevel logistic models consisting of the province, community, and individual levels. The key findings and their implications are summarized as follows.

First, we obtained evidence of associations between vaccination and general trust and interaction with neighbors, but not with trust in government, with variations between the age groups. Specifically, the positive correlation between vaccination and the levels of general trust among the young population is in line with previous studies that focused on social capital [10,11,12]. Similarly, the positive relationship between vaccination and interaction with neighbors is consistent with the general trust results, as a high level of interaction with neighbors reflects close and intimate relationships with neighbors, which is a key element of neighborhood social capital [17,18]. These findings are noteworthy, considering that a high proportion (nearly 90%) of participants were vaccinated and that we controlled for the possibility of political or institutional pressures on vaccine uptake in the regression analysis.

The discrepancy between our findings on trust in government and previous studies [13,14,15] may be attributed to a lack of variation in the levels of trust in government within our study sample. Approximately 91% of young adults and 90% of older adults reported high levels of trust in government. This high rate might be due to face-to-face interview data collection. The previous studies collected data through online surveys or via mail or text messages and showed a large variation in the levels of trust in government among the respondents [13,14,15]. Future studies may consider employing more private methods for collecting such information to avoid potential report bias and examine the role of trust in government among the Chinese population.

Second, we found that vaccination was positively associated with community-level general trust and interaction with neighbors among the young population, even after controlling for individual-level measures. Additionally, there is evidence for community-level interaction with neighbors among older adults without controlling for individual-level measures. Although this study did not examine interactions or causation between individual- and community-level measures, the results suggest that living in a community with high levels of general trust or interaction with neighbors may enhance vaccination rates among young adults, even after controlling for an individual’s views and behaviors. This finding is consistent with previous studies that emphasized the importance of neighborhood social capital for an individual’s well-being [19,20,21,22,23].

In the context of China, the community plays a crucial role in local neighborhood self-governance. In early 2020, the central leadership of China issued a series of government regulations for epidemic containment, and it falls upon the neighborhood committees to enforce these regulations at the community level [33]. Throughout the COVID-19 outbreak in China, these committees played a pivotal role in epidemic preparedness, control, and the implementation of lockdown measures. Given the significance of community in China’s grassroots self-governance, it is not surprising to observe a significant association between the proxies of community-level social capital and high vaccination rates.

Third, we uncovered age differences in the associations between vaccination and general trust, trust in government, and interaction with neighbors. Particularly noteworthy is the finding that among older adults, who are considered more susceptible to COVID-19 infection and tend to experience more severe consequences if they contract the SARS-CoV-2 virus, a higher level of interaction with neighbors was associated with a higher vaccination rate. However, this pattern was not observed among the young population, for whom general trust at both the individual and community levels played a more prominent role. A previous study also found a beneficial role of neighborhood social cohesion in improving willingness to receive a booster dose of COVID-19 vaccines among the older Chinese population [34]. Conversely, neighborhood disorder has been found to be associated with a lower level of preventive healthcare utilization [35]. The positive association between interaction with neighbors and vaccination may be explained by enhanced neighborhood communication and knowledge exchange, serving as a means of peer health education among neighbors, especially for older adults with limited social networks [34]. Even when older individuals live with their adult children, they may still spend considerable free time with neighbors within the community [36].

The current cohort of the older population in China is characterized by limited educational levels [36]. This makes interaction with neighbors more instrumental in health promotion. Neighbors can play a significant role in communicating health information and serve as role models for behavior change. A previous study found that interactions with neighbors were instrumental in promoting health literacy among Chinese speakers in the USA who had limited English proficiency [36]. Another study showed that a higher level of social support from neighbors was associated with improved health literacy among older adults during the COVID-19 pandemic [37]. In our study sample, only 37.5% of older respondents had a high level of interaction with neighbors. Although our measure of interactions with neighbors only captured physical interactions, virtual interactions may be more prevalent during the pandemic when social distancing was implemented. However, evidence shows that many older adults in China were unable to use mobile technologies during the pandemic [38]. Given the relatively low vaccination rate among older adults in our study sample, an increase in neighborhood interactions could have potentially improved vaccine uptake.

A better understanding of the role of neighborhood interactions for older people in vaccine uptake can assist in the development and implementation of immunization campaigns. Promoting interaction within the neighborhood, along with effective risk communication and health education among community residents, holds great promise in combating misinformation or conspiracy theories against vaccines, enhancing health literacy, promoting preventive health behavior, and increasing vaccine uptake. Moreover, even when COVID-19 is no longer a public health emergency, the importance of neighborhood interactions for older adults remains relevant. Our findings have significant public health implications for other diseases to which older people are susceptible, such as seasonal influenza, where evidence indicates a low vaccination rate among the older Chinese population [39]. Furthermore, it is worthwhile to investigate the effects of different modalities (physical and virtual) of interactions with neighbors on vaccine uptake, as the COVID-19 experience taught us the importance of digital devices for older adults to interact with others during times of social isolation.

This study has several limitations. First, like many cross-sectional studies, we cannot rule out the possibility of reverse causation, meaning that the outcome may have affected the explanatory variables. For instance, it is possible that vaccine uptake, together with neighbors’ vaccine uptake and changes in infection rates in the region, may have affected interaction with neighbors and even general trust. In addition, trust in government levels is likely to improve if increasing vaccination successfully contains the infection.

Second, since the 2021 CGSS was conducted using face-to-face interviews, there might be a response bias when participants were asked about their attitudes toward the government.

Third, our measurement of interaction with neighbors only accounted for physical interaction. During pandemic outbreaks when movement was restricted, virtual interaction with neighbors may also have a substantial impact on individual vaccine uptake.

Fourth, due to a lack of longitudinal data, we were unable to analyze vaccination dynamics, which could provide valuable insights for future public health practices. Vaccination is likely to be affected by and also have an impact on others’ behavior, and this interaction is expected to be shaped by general trust, trust in government, and interaction with neighbors. Moreover, these factors may also be influenced by the dynamic immunization behaviors among individuals.

Finally, caution should be exercised when generalizing the results of this study, as our primary focus was on China, which has its unique contextual characteristics, such as a remarkably high vaccination rate and the presence of suggested political or institutional pressures on vaccine uptake based on our findings.

## 5. Conclusions

The results of this study showed that vaccine uptake was associated with an individual’s views of society and the social environment within their community, with substantial variations between the young and the old. Notably, the association between interaction with neighbors and vaccination rates among older adults, who had relatively low vaccination rates, emphasizes the significance of public health measures to strengthen neighborhood communication and exchange. This approach can take advantage of peer education and support to better protect the susceptible population against COVID-19 or other infectious diseases where older people are at high risk.

## Figures and Tables

**Table 1 vaccines-11-01332-t001:** Key characteristics of the study sample.

Proportion (%)		All	Men	Women
Marital status				
Married		74.0	75.4	72.3
Unmarried		15.1	12.3	18.5
Divorced/separated		10.9	12.3	9.2
Educational attainment				
Illiterate		9.1	12.6	4.9
Primary school		20.8	21.9	19.6
Junior high school		28.8	28.2	29.5
High school		18.8	16.2	22.0
College or above		22.5	21.1	24.1
Communist party member		19.8	15.6	24.8
Poor self-rated health		14.9	16.3	13.3
Living alone		11.3	10.6	12.2
Agricultural hukou		59.9	61.5	58.0
Occupation type				
No work		45.4	52.4	37.1
Farming		16.1	14.9	17.5
Government-related work ^a^		11.1	10.0	12.5
Private or foreign company		12.7	10.6	15.2
Self-employed		12.7	10.3	15.6
Other		1.9	1.7	2.1
Age (years)	M	48.6	48.1	49.0
	SD	(16.9)	(16.4)	(17.4)
Family income	M	66.8	59.3	75.4
(annual, equivalized, 1000 CNY)	SD	(238.0)	(180.7)	(289.8)
*N*		6860	3730	3130

^a^ Encompassed works in (1) party and government institutions; (2) state-owned or collectively owned enterprises, business groups, social groups, neighborhood or village committees; and (3) the army.

**Table 2 vaccines-11-01332-t002:** Vaccination rates (%) by sex and age.

Age	Men (*N* = 3730)	Women (*N* = 3130)	All (*N* = 6860)
18–29	94.5	84.0	89.5
30–39	92.3	82.8	88.4
40–49	93.6	92.6	93.2
50–59	91.7	90.9	91.4
60–69	84.0	81.5	82.8
70–79	74.3	76.5	75.5
80+	50.7	59.0	54.7
Total	89.2	84.8	87.2

**Table 3 vaccines-11-01332-t003:** Vaccination rates by individual-level trust and interaction with neighbors.

	*N*. of Individuals		Vaccination Rate (%)			
	High	Low	High	Low	Difference	
			(A)	(B)	(A)—(B)	95% CI
Young (*N* = 4859)						
General trust	3204	1655	92.0	88.2	3.8	(2.0, 5.6)
Trust in government	4427	432	90.9	89.1	1.8	(–1.3, 4.8)
Interaction with neighbors	1261	3598	92.7	90.0	2.7	(0.9, 4.4)
Old (*N* = 2001)						
General trust	1508	493	79.4	76.5	2.9	(–1.4, 7.2)
Trust in government	1792	209	79.4	72.2	7.2	(0.8, 13.6)
Interaction with neighbors	750	1251	81.7	76.8	4.9	(1.3, 8.5)

**Table 4 vaccines-11-01332-t004:** Estimated associations of trust and interaction with neighbors with vaccination rate: multilevel logistic models ^a^.

	Model 1			Model 2			Model 3		
	OR	95% CI		OR	95% CI		OR	95% CI	
Young (*N* = 4859)									
General trust									
Individual-level	**1.41**	(1.13, 1.77)					**1.35**	(1.07, 1.70)	
Community-level				**1.68**	(1.21, 2.31)		**1.58**	(1.14, 2.18)	
Trust in government									
Individual-level	1.41	(0.98, 2.04)					**1.46**	(1.00, 2.12)	
Community-level				0.88	(0.63, 1.21)		0.84	(0.60, 1.16)	
Interaction with neighbors									
Individual-level	1.19	(0.91, 1.57)					1.13	(0.85, 1.49)	
Community-level				**1.59**	(1.14, 2.21)		**1.55**	(1.11, 2.17)	
Old (*N* = 2001)									
General trust									
Individual-level	1.23	(0.91, 1.66)					1.21	(0.90, 1.64)	
Community-level				1.20	(0.80, 1.79)		1.17	(0.78, 1.75)	
Trust in government									
Individual-level	1.35	(0.89, 2.05)					1.34	(0.88, 2.04)	
Community-level				1.09	(0.74, 1.60)		1.05	(0.71, 1.55)	
Interaction with neighbors									
Individual-level	**1.63**	(1.22, 2.17)					**1.55**	(1.15, 2.08)	
Community-level				**1.63**	(1.08, 2.46)		1.44	(0.94, 2.20)	

^a^ All three models controlled for a set of individual covariates (sex, age, marital status, living alone, educational attainment, occupation, family income level, poor self-rated health, hukou type, and being a Communist Party member) and two province-level covariates (levels of new COVID-19 cases per capita and vaccination rates). See Appendix A for the detailed results in the case of general trust. Bold indicates *p* < 0.05.

## Data Availability

The CGSS data used in this study can be downloaded from http://www.cnsda.org/index.php?r=projects/view&id=62072446 (accessed on 11 July 2023)).

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
