# Peer review of "Trust, Interaction with Neighbors, and Vaccination during the COVID-19 Pandemic: A Cross-Sectional Analysis of Chinese Data"

_vaccines, 2023, doi:10.3390/vaccines11081332_

Round 1

Reviewer 1 Report

I read the study by Takashi Oshio and Ping Ruru entitled "Trust, interaction with neighbors, and vaccination during the COVID-19 pandemic: a multilevel analysis in China". In summary I would argue that it is a publishable manuscript. However a dark point is the conversion of variables to binary, the authors should explain why they chose to modify them.

Here are my observations:

- It is necessary to increase the references in both the introduction and the discussion.

-Please provide more information on the exclusion of 1288 individuals who did not meet vaccination requirements.

-Line 184 delete the word "Result"

Author Response

We appreciate your constructive comments. In what follows, we explain how we have responded to them. Line numbers are for the revised manuscript.

I read the study by Takashi Oshio and Ping Ruru entitled "Trust, interaction with neighbors, and vaccination during the COVID-19 pandemic: a multilevel analysis in China". In summary I would argue that it is a publishable manuscript. However a dark point is the conversion of variables to binary, the authors should explain why they chose to modify them.

=> Thank you for your positive evaluation. Regarding your comment on binary variables, we added the explanation: “We constructed a binary variable for each construct, because it is difficult to assess it by a continuous measure. Moreover, a binary variable can help straightforwardly assess the magnitude of its association with vaccine uptake” (lines 121–124).

Here are my observations:

- It is necessary to increase the references in both the introduction and the discussion.

=> We added new nine references to Introduction and the Discussion; in particular, those that discussed age differences and their implications, on which the revise version put more emphasis. We also added missing references in Introduction (line 37, 55).

-Please provide more information on the exclusion of 1288 individuals who did not meet vaccination requirements.

=> We added the explanation: “The CGSS questionnaire asked about the reasons of non-vaccination, which included an option of not meeting vaccination requirements. We used this information as an exclusion criterion for our study sample” (Lines 105-108).

-Line 184 delete the word "Result"

=> We removed the word.

Reviewer 2 Report

1-      Abstract should be formatted according to the Journal format

2-      Conceptualizing and measuring main constructs of general trust in others (GT), trust in government (TG), and interaction with neighbors (IN) have not been well done

3-      The culturally and theoretically reasons of Gender differences analyses have not justified

4-      Please provide theoretical and practical implications of the study

5-      The analyses have included only vaccinated samples, it seems that all samples be included and compared  

Author Response

We appreciate your constructive comments. In what follows, we explain how we have responded to them. Line numbers are for the revised manuscript.

1-Abstract should be formatted according to the Journal format

=> We reformat the Abstract in accordance with the journal’s formatting requirement.

2-Conceptualizing and measuring main constructs of general trust in others (GT), trust in government (TG), and interaction with neighbors (IN) have not been well done

=> We provided a more detailed discussion about the concepts of the three main constructs and relevance to this study: “General trust in others is a key element of social capital, which is defined as the quality of relationships among community members [16]. Previous studies often use general trust an indicator of social capital to examine its association with vaccine hesitancy [10–12]. Higher general trust is expected to increase individuals’ concern for others and engagement in practices that aim to improve the overall situation.

Trust in government, which refers to people’s beliefs that the government acts transparently and fairly in accordance with the public interest, has also been investigated as a factor related to vaccination willingness [13–15]. Analyzing vaccination practices in the context where government takes the lead in managing the pandemic crisis makes it even more important to consider trust in government.

As a correlate of vaccination, we also focused on interaction with neighbors, which is considered a vital component of neighborhood-based social capital [17,18]. More frequent and close interactions with neighbors are expected to help individuals share pandemic-related information and increase support for neighbors, thereby promoting vaccine uptake” (Lines 40-54). We also explicitly explained in Measures section how we incorporated these three variables in our analysis (lines 125-148).

3-The culturally and theoretically reasons of Gender differences analyses have not justified

=> Thank you very much for your constructive comment. We have replaced the focus on gender difference, which we found is not much relevant in this study, with an emphasis on age difference and provided our rationale for conducting this analysis as follows: “Additionally, we investigated how the association between vaccine uptake and general trust, trust in government, and interaction with neighbors differed between young and old populations. This investigation was prompted by our observation of a low vaccination rate among older respondents in our study sample, which is consistent with findings from other studies” (lines 77–81).

4-Please provide theoretical and practical implications of the study

=> We included a new paragraph in the discussion to outline the public health implications of our findings: “A better understanding of the role of neighborhood interactions for older people in vaccine uptake can assist in the development and implementation of immunization campaigns. Promoting interaction within the neighborhood, along with effective risk communication and health education among community residents, holds great promises in combating misinformation or conspiracy theories against vaccines, enhancing health literacy, promoting preventive health behavior, and increasing vaccine uptake. Moreover, even when COVID-19 is no longer a public health emergency, the importance of neighborhood interactions for older adults remains relevant. Our findings have significant public health implications for other diseases to which older people are susceptible, such as seasonal influenza, where evidence indicates a low vaccination rate among the Chinese older population. Furthermore, it is worthwhile to investigate the effects of different modalities (physical and virtual) of interactions with neighbors on vaccine uptake, as the COVID-19 experience taught us the importance of digital devices for older adults to interact with others during times of social isolation” (lines 348-361).

5-The analyses have included only vaccinated samples, it seems that all samples be included and compared

=> We analyzed both vaccinated and non-vaccinated individuals. We excluded only those who were not qualified to get vaccinated, as mentioned in study sample section (lines 103–105).

Reviewer 3 Report

This study with a very large N is well described with a relatively complex statistical analysis.

I've made my comments directly in the PDF for a better reading of the article by the reader.

Author Response

We appreciate your constructive comments. In what follows, we explain how we have responded to them. Line numbers are for the revised manuscript.

Page 1

Add in the title that this is a cross-sectional study

=> Following your suggestion, we revised the title to “Trust, interaction with neighbors, and vaccination during the COVID-19 pandemic: a cross-sectional analysis of Chinese data.”

You need the purpose of the study at the end of the background

=> We removed subtitles (such as Background, Method) in accordance with the journal’s guideline (as requested by another reviewer), and hence the purpose of the study now is stated in the second sentence.

page 2

The aim of the study should be placed in the last chapter.

=> We reconstructed Introduction to place the aim of the study at its end. See lines 67-84.

There is a lack of information about this CGSS, more information is needed

=>We added the explanation about the CGSS: “Initiated in 2003, the CGSS was the first nationwide and continuous large-scale social survey project in China. The CGSS aims to collect data on family structure, education, employment, health, social attitudes, and other important topics in order to monitor changes in Chinese society over time and investigate social issues with theoretical and practical significance. The China Survey and Data Center of Renmin University of China is responsible for the implementation, management, and data release of the project. The CGSS adopted a multi-stage stratified sampling approach. In CGSS, counties served as primary sampling units, followed by urban communities and rural villages as secondary sampling units, and households were randomly selected using a mapping sampling approach [28]” with a new reference [28] for more detailed information (lines 87-96). This is followed by details on survey designs and other information.

page 6

Statistically significant ORs should be highlighted in bold.

=> We followed your suggestion and added “Bold indicates p < .05.” to the table’s footnote.

Reviewer 4 Report

This manuscript report results of a multilevel analysis of Chinese vaccination data. Although the overall topic is relevant and the data procedures used seem overall plausible, the article has several shortcomings.

The title should be modified to be clearer, such as “… analysis of Chinese data”.

The abbreviations used in the abstract and in the main text limit reading flow and should be avoided for uncommon abbreviations.

The rationale behind the study is not clearly described, as it is not explained how exceptionally high vaccination rates can provide valuable insights for public health authorities to better design and manage immunization campaigns in the future. From the results, the only group with low vaccination rates are the elderly. Unfortunately, this finding is not further evaluated or commented on in greater detail.

There are several typos, formatting issues, and missing references throughout the text. The cited websites should be transformed into references.

In sum, the practical and theoretical findings of the results are not sufficiently outlined. Also, the overall merit for an international readership is quite low, as the country-specificity of the results limit generalizability of the results.

There are several typos and formatting issues that should be corrected.

Author Response

We appreciate your constructive comments. In what follows, we explain how we have responded to them. Line numbers are for the revised manuscript.

This manuscript report results of a multilevel analysis of Chinese vaccination data. Although the overall topic is relevant and the data procedures used seem overall plausible, the article has several shortcomings.

The title should be modified to be clearer, such as “… analysis of Chinese data”.

=> We modified the title to: “Trust, interaction with neighbors, and vaccination during the COVID-19 pandemic: a cross-sectional analysis of Chinese data.”

The abbreviations used in the abstract and in the main text limit reading flow and should be avoided for uncommon abbreviations.

=> We removed the uncommon abbreviations (GT, TG, IN) in abstract and main text.

The rationale behind the study is not clearly described, as it is not explained how exceptionally high vaccination rates can provide valuable insights for public health authorities to better design and manage immunization campaigns in the future. From the results, the only group with low vaccination rates are the elderly. Unfortunately, this finding is not further evaluated or commented on in greater detail.

=> Thank you very much for your constructive comment. We have replaced the focus on gender difference with an emphasis on age difference and provided our rationale for conducting this analysis, incorporating your valuable insights, as follows: “Additionally, we investigated how the association between vaccine uptake and general trust, trust in government, and interaction with neighbors differed between young and older populations. This investigation was prompted by our observation of a low vaccination rate among older respondents in our study sample, which is consistent with findings from other studies” (lines 77–84).

Furthermore, we updated our findings to highlight age difference in Table 3–4 and Table S2, as well as in result section (lines 217-269). Accordingly, we provided our interpretations of the updated findings in discussion section (lines 279–362), with a particular focus on age differences (lines 317–361).

There are several typos, formatting issues, and missing references throughout the text. The cited websites should be transformed into references

=> We have made corrections as you suggested.

In sum, the practical and theoretical findings of the results are not sufficiently outlined. Also, the overall merit for an international readership is quite low, as the country-specificity of the results limit generalizability of the results.

=> We included a new paragraph in the discussion to outline the practical implications of our findings: “A better understanding of the role of neighborhood interactions for older people in vaccine uptake can assist in the development and implementation of immunization campaigns. Promoting interaction within the neighborhood, along with effective risk communication and health education among community residents, holds great promises in combating misinformation or conspiracy theories against vaccines, enhancing health literacy, promoting preventive health behavior, and increasing vaccine uptake. Moreover, even when COVID-19 is no longer a public health emergency, the importance of neighborhood interactions for older adults remains relevant. Our findings have significant public health implications for other diseases to which older people are susceptible, such as seasonal influenza, where evidence indicates a low vaccination rate among the Chinese older population. Furthermore, it is worthwhile to investigate the effects of different modalities (physical and virtual) of interactions with neighbors on vaccine uptake, as the COVID-19 experience taught us the importance of digital devices for older adults to interact with others during times of social isolation” (lines 348-361).

Regarding your second point, we mentioned country-specificity as the fifth limitation in Discussion (lines 380-385).

Reviewer 5 Report

Dear Authors

I found your study interesting, despite the limitations of the study, offering an interesting picture of the reality of better sizing policies that enhance vaccination.

Author Response

I found your study interesting, despite the limitations of the study, offering an interesting picture of the reality of better sizing policies that enhance vaccination.

=> We thank you for your positive evaluation. We have thoroughly revised the manuscript in response to all reviewers’ comments and suggestions.

Round 2

Reviewer 2 Report

Thanks for considering the comments 

Reviewer 4 Report

No further revisions needed

 Minor editing of English language required